# Differential Immune Checkpoint and Ig-like V-Type Receptor Profiles in COVID-19: Associations with Severity and Treatment

**DOI:** 10.3390/jcm11123287

**Published:** 2022-06-08

**Authors:** Roberto Lozano-Rodríguez, Verónica Terrón-Arcos, Raúl López, Juan Martín-Gutiérrez, Alejandro Martín-Quirós, Charbel Maroun-Eid, Elena Muñoz del Val, Carlos Cañada-Illana, Alejandro Pascual Iglesias, Jaime Valentín Quiroga, Karla Montalbán-Hernández, José Carlos Casalvilla-Dueñas, Miguel A. García-Garrido, Álvaro del Balzo-Castillo, María A. Peinado-Quesada, Laura Gómez-Lage, Carmen Herrero-Benito, Ray G. Butler, José Avendaño-Ortiz, Eduardo López-Collazo

**Affiliations:** 1The Innate Immune Response Group, IdiPAZ, La Paz University Hospital, Paseo de la Castellana 261, 28046 Madrid, Spain; roberto.lozano.rodriguez@idipaz.es (R.L.-R.); vterronarcos@gmail.com (V.T.-A.); alejandro.pascual.iglesias@idipaz.es (A.P.I.); jaimevquiroga@hotmail.com (J.V.Q.); karlamarina.hernandez@gmail.com (K.M.-H.); jccasalvilla@yahoo.com (J.C.C.-D.); abalzcas@myuax.com (Á.d.B.-C.); 2Tumor ImmunologyLaboratory, IdiPAZ, La Paz University Hospital, Paseo de la Castellana 261, 28046 Madrid, Spain; 3Butler Scientifics S.L., 08035 Barcelona, Spain; raulguti7@gmail.com (R.L.); juan@butlerscientifics.com (J.M.-G.); ray@butlerscientifics.com (R.G.B.); 4Emergency Department and Emergent Pathology Research Group, IdiPAZ La Paz University Hospital, Paseo de la Castellana 261, 28046 Madrid, Spain; a.martinquiros@gmail.com (A.M.-Q.); charbel.maouneid@gmail.com (C.M.-E.); elena.munoz.dv@gmail.com (E.M.d.V.); carlos.canada.illana@idipaz.es (C.C.-I.); miguelangelgarciagarrido@gmail.com (M.A.G.-G.); mariapq1@gmail.com (M.A.P.-Q.); lauragomezlage@gmail.com (L.G.-L.); c.herrerobenito@gmail.com (C.H.-B.); 5CIBER of Respiratory Diseases (CIBERES), 28029 Madrid, Spain

**Keywords:** COVID-19, immune phenotype, immune-checkpoints, Ig-like V-type receptors, secondary infections, corticosteroids

## Abstract

Identifying patients’ immune system status has become critical to managing SARS-CoV-2 infection and avoiding the appearance of secondary infections during a hospital stay. Despite the high volume of research, robust severity and outcome markers are still lacking in COVID-19. We recruited 87 COVID-19 patients and analyzed, by unbiased automated software, 356 parameters at baseline emergency department admission including: high depth immune phenotyping and immune checkpoint expression by spectral flow cytometry, cytokines and other soluble molecules in plasma as well as routine clinical variables. We identified 69 baseline alterations in the expression of immune checkpoints, Ig-like V type receptors and other immune population markers associated with severity (O_2_ requirement). Thirty-four changes in these markers/populations were associated with secondary infection appearance. In addition, through a longitudinal sample collection, we described the changes which take place in the immune system of COVID-19 patients during secondary infections and in response to corticosteroid treatment. Our study provides information about immune checkpoint molecules and other less-studied receptors with Ig-like V-type domains such as CD108, CD226, HVEM (CD270), B7H3 (CD276), B7H5 (VISTA) and GITR (CD357), defining these as novel interesting molecules in severe and corticosteroids-treated acute infections.

## 1. Introduction

The emergence of severe acute respiratory syndrome coronavirus 2 (SARS-CoV-2) and its associated disease (coronavirus disease 2019, COVID-19) has had a strong impact on societies worldwide. The introduction of COVID-19 vaccines has vastly decreased severe COVID-19 cases. However, the arrival of new strains and the exponential increase in cases, continue to make this disease a problem in many countries. The only treatment that showed an improvement in patients’ survival during the first waves of the pandemic was the use of corticosteroids [1,2]. Nonetheless, critically ill and drug-immunosuppressed patients are highly susceptible to develop secondary bacterial infections such as secondary pneumonia, bloodstream infections and catheter-related sepsis, all of which strongly impact the final outcome [3,4,5,6,7,8]. These data highlight the need for continued study of the immune response in this pathology.

Since the World Health Organization (WHO) declared COVID-19 a pandemic, the disease has been associated with the “cytokine storm” phenomenon; however, other immunological processes such as lymphocytopenia, T-cell depletion and defects in the innate immune system have also been observed [9,10,11]. Beyond the changes in cell population’s profile, changes in activation and inhibitory receptors have also been described in some immune populations in the course of SARS-CoV-2 infection [11,12,13,14,15]. Within these receptors, the B7 superfamily and immune-checkpoints (ICs) stand out for their role in modulating immune response and their involvement in cell-to-cell communication [16,17,18]. This type of molecule is characterized by having an immunoglobulin (Ig)-like V-type domain by which they interact with each other via ligand and receptor [19,20]. We and others have described increased soluble levels of some ICs such as T cell immunoglobulin and mucin-domain containing-3 (Tim-3) and programmed death-ligand 1 (PD-L1) in severe COVID-19 patients’ plasma, establishing these as better severity markers than cytokines [13,21,22,23]. In addition, the use of blocking monoclonal antibodies against these kinds of molecules has been successfully implemented as immunotherapy in oncology [24,25]. For this reason, the study of the immune cells’ phenotype distribution and its immune checkpoint expression throughout the infection course remains relevant.

In this report, we have collected longitudinal samples from 87 COVID-19 patients from emergency department admission until exitus or discharge. We have performed an unbiased multiparametric analysis including clinical data, levels of soluble molecules in plasma (cytokines, chemokines, ICs and thrombosis markers) and data from two complex spectral cytometry panels: a panel for detailed cell subpopulation description and a panel for the study of 21 activation markers, including immune checkpoints and Ig-like V-type receptors, in the main immune populations. Note that, spectral cytometry enables the simultaneous study of up to 40 immunological markers, exponentially increasing the complexity and detail of the data obtained [26,27]. Our analysis has provided an exhaustive description of the immune system in patients classified according to two of the major COVID-19 complications: the appearance of secondary infections and O_2_ requirement during their stay in hospital. In addition, we have analyzed the variations of immunological parameters after secondary infections and the changes due to corticosteroid treatment. Altogether, our data sheds light on new potential severity markers at admission as well as a personalized patient management.

## 2. Materials and Methods

### 2.1. Patient Recruitment and Study Design

We designed a prospective observational study in which we recruited 87 patients from 26 April 2020 to 20 November 2020 during the first two waves of COVID-19 in Madrid, Spain. Due to the descriptive characteristic of our study, there was no predefined sample size. Patients were included when they met the diagnostic criteria for COVID-19 and were positive for SARS-CoV-2 by real-time quantitative polymerase chain reaction (RT-qPCR) from nasopharyngeal swabs. Patients were recruited consecutively at the emergency department (ED) when the clinical investigation team was available, independently of the patients’ clinical status and prior to their hospitalization in La Paz University Hospital in Madrid (Spain). Patients receiving immunosuppressants at admission (i.e., chemotherapy or corticosteroids) and those with immunodeficiency (primary or acquired) were excluded from the study. For each patient, we took a sample on admission and previously to any treatment. Longitudinal samples every 2–4 days until discharge or exitus were collected. Mortality endpoint in exitus patients was defined at 60 days after admission. Those patients with a positive test in the culture of samples such as blood, urine, bronchoalveolar lavage fluid (BALF) or stool during hospital admission were identified as secondary infected patients. Corticosteroid treatment consisted of intravenous dexamethasone 8 mg daily or oral dexamethasone 6 mg daily (until exitus or discharge, mean treatment duration: 4.7 days) [2]. Participants signed an informed consent and data were anonymized before study inclusion. All parameters included in the study are described at Appendix A.

### 2.2. Immunophenotyping

Fresh blood from venipuncture was collected in Lithium heparin and K_2_ ethylenediaminetetraacetic acid (EDTA) anticoagulant tubes (Vacuette^®^, Greiner Bio One, Kremsmünster, Austria). Whole blood cells were obtained after red blood cells lysis. Briefly, 2 mL of heparinized blood were treated with 20 mL of 1X Pharm Lyse Buffer (BD Biosciences, San Jose, CA, USA) for 15 min at room temperature (RT) in a rocker. Then, cells were washed twice in PBS and stained with LIVE/DEAD Fixable Blue Dead Cell Stain Kit (Invitrogen, Vienna, Austria) for 15 min to discard dead cells. True-Stain Monocyte Blocker (BioLegend, San Diego, CA, USA) reagent was added prior to labeling to block the nonspecific binding of some fluorochromes to monocytes. After that, the cells were labeled for 25 min at RT in the dark with fluorochrome-conjugated monoclonal antibodies. The labelling consisted in two cytometry panels (Appendix A) to analyze immune cell populations and immune-checkpoint expression respectively. Labeled cells were acquired on a Cytek Aurora Spectral Cytometer (Cytek Biosciences, Bethesda, MD, USA). Data were analyzed using FlowJo (TreeStar, Ashland, OR, USA) v10.6.2 software. The gating strategy followed is shown in Appendix A.

### 2.3. Plasma and PBMCs Isolation

Peripheral blood mononuclear cells (PBMCs) and plasma were isolated from fresh blood of K_2_EDTA tubes by Ficoll-Plus (GE Healthcare Bio-Sciences, Piscataway, NJ, USA) gradient, following manufacturer’s instructions. Plasma was stored at −80 °C until the analysis.

### 2.4. Cytokines, Soluble Immune-Checkpoints and Thrombosis Markers Quantification

Quantification of soluble markers including cytokines, soluble immune-checkpoints and thrombosis markers was performed by cytometric bead array (LEGENDplex Human Essential Immune response panel, LEGENDplex HU Immune Checkpoint Panel 1 and LEGENDplex™ Human Thrombosis Panel, respectively, all three purchased from Biolegend) according to the manufacturer’s instructions. Samples were acquired on a FACSCalibur flow cytometer (BD Biosciences, San Jose, CA, USA) and data were analyzed using the LEGENDplex™ Data Analysis Software Suite (Qognit, Inc., San Carlos, CA, USA).

### 2.5. Statistical Analysis

We included a total of 356 variables from COVID-19 patients (Appendix A). Exploratory data were analyzed in an unbiased automated manner by AutoDiscovery software (Butler Scientifics, Barcelona, Spain). The variance analyses were performed by three methods: one-way ANOVA, when the response fitted a normal distribution (Jarque-Bera test); Mann–Whitney U, when the response did not fit a normal distribution and factors had exactly two categories; and Kruskal–Wallis, when the response did not fit a normal distribution and the factors had more than two categories. For qualitative variables, Cramer’s V Contingency Index was used. For longitudinal analysis of changes due to secondary infection, the levels of immunological parameters from previous samples and the samples after the day of culture-positive sample collection were analyzed by Mann–Whitney test. The same test was used to study the changes in response to corticosteroid treatment (mean treatment duration: 4.7 days). Levels of immunological parameters in samples previous-to-treatment were compared with the levels in the samples after treatment initiation. Given the multiple-testing nature of the analysis, significance was adjusted using a false-discovery rate (FDR) method (Benjamini–Hochberg). Adjusted *p*-value thresholds ranged from 0.002 to 0.00003 depending on the analysis stage. Results were then classified in three levels based on the following significance thresholds: (i) exploratory results (FDR-threshold < *p*-value < 0.05), (ii) high-significance/confirmatory results (*p*-value < FDR-threshold) and (iii) rejected results (*p*-value >= 0.05). Only exploratory and high-significance results were included in the manuscript. Additionally, subgroups or associations with a sample size below 5 or a sample size below 1% of the total sample size were automatically rejected.

## 3. Results

### 3.1. Parameters at Admission Associated to COVID-19 Severity

One of the key actions in the COVID-19 pandemic is to quickly identify patients who will progress poorly before they get extremely ill. In order to define severity markers, we recruited 87 patients at emergency department (ED) admission, and analyzed 356 baseline parameters including clinical characteristics, routine laboratory markers, cytokines, chemokines, immune-checkpoints and thrombosis markers and immune phenotype (Appendix A). All patients were followed up until discharge or exitus and classified into five groups according to their hospital requirements during their stay and final outcome: 1, no hospital requirement (*n* = 22); 2, hospitalized with no O_2_ requirement (*n* = 10); 3, hospitalized with O_2_ requirement (*n* = 23); 4, orotracheal intubation (OTI) requirement (*n* = 13); and 5, exitus patients (*n* = 19). Due to the vast amount of data, differences between groups were analyzed by an automatic bioinformatic analysis. Those variables with statistically significant differences are described below.

#### 3.1.1. Patients’ Clinical Characteristics and Routine Laboratory Markers

We found significant differences between the five severity groups in 23 parameters of clinical characteristics and routine laboratory data at admission (Appendix A). As expected, those parameters included age, quick-Sequential Organ Failure Assessment (qSOFA), respiratory parameters, inflammation-related markers, and changes in leukocytes distribution. Among inflammatory markers, is noteworthy the C-reactive protein (CRP) and Ferritin characterized by higher levels in patients with OTI requirement or who subsequently died. In terms of leukocyte counts, OTI-required and exitus patients were characterized by a marked lymphocytopenia and neutrophilia. It is also relevant to point out the high levels of D-dimer and low platelet counts in severely ill patients, which is in accordance with the prothrombotic events described in these patients [28,29,30].

#### 3.1.2. Soluble Plasma Markers

Since inflammatory markers, leukocytes and thrombosis parameters are acutely altered in severe patients (OTI-required and exitus) at admission, we chose to study cytokines, soluble ICs and thrombosis markers to give a more detailed profile of COVID-19 patients. We found CLL2 and CXCL10 chemokines as well as cytokines IL-6, IL-8 and IL-10 showed higher levels in exitus patients (Appendix A). As we have previously reported, the ICs: sCD25, sTim-3 and Galectin-9 (Gal-9) were increased according to severity [13,22]. Likewise, two thrombosis-related markers, the tissue plasminogen activator (tPA) and the P-Selectin, were upregulated in severe and O_2_-required patients.

#### 3.1.3. Immune Cell Subpopulations and Their Immune Checkpoint Expression

Once the leukocytes unbalance and the high soluble IC levels in severe COVID-19 patients were established, we proceeded to study the associations of severity within immune cell subpopulations and their membrane immune checkpoint expression. We found statistically significant differences between groups in monocytes, neutrophils, dendritic cells (DCs), natural killer (NK) cells, T cells, γδ^+^ cells and B cell populations (Table 1).

Monocyte population in severe patients (both exitus and OTI-required patients) were characterized by increment of myeloid-derived suppressor cells (m-MDSCs), higher expression of CD16, CD206 and PD-L1 and lower expression of human leukocyte antigen (HLA)-DR. In addition, exitus patients showed the highest expression of CD62L in all monocyte subpopulations (classical, intermediate and non-classical). Regarding neutrophils, exitus patients had higher expression of CD206, CD226 and Tim-3. Other innate immune cells analyzed were the DCs, showing lower frequency of myeloid DCs and CD141^+^ cells in severe patients, and the NK cells, characterized by higher expression of CD108, CD162 and CD321 in exitus patients. Other ICs such as glucocorticoid-induced TNFR-related (GITR) or T cell immunoglobulin and ITIM domain (TIGIT) did not exhibit a clear pattern (Table 1). The adaptive immune system of severe patients was characterized by increased expression of CD162 and CD223 and lower expression of CD226 in all CD3^+^, CD4^+^ and CD8^+^ T cells. The CD4^+^ T cells showed an increase in effector memory and terminally differentiated subpopulations in patients who subsequently died. Note that, CD8^+^ T cells showed higher expression of the proapoptotic receptor CD95 as well as enhanced expression of GITR in severe patients (Table 1). Other adaptive immune cell populations such as γδ^+^ T cells, CD19^+^ B cells or transitional B cells, showed decreased frequency in severe patients.

### 3.2. Parameters at Admission Associated with Secondary Infections in COVID-19 Patients

Beyond severity due to oxygen requirement during hospitalization, the appearance of secondary infections remains a major issue in the management of COVID-19 patients [3,4,8,31]. We compared the baseline parameters, classifying patients according to whether they developed secondary infections during their stay. Thirty-three of the 87 patients developed one or more co-infections confirmed by a positive test in the culture of samples such as blood, urine, bronchoalveolar lavage fluid (BALF) or stool.

#### 3.2.1. Patients’ Clinical Characteristics and Routine Laboratory Markers at Admission Associated with Secondary Infections

In contrast to what we observed when classified according to severity, we did not find differences in age between those who suffered secondary infection and those who did not. Instead, we found differences in sex, with a higher frequency of men among COVID-19 patients who subsequently suffered secondary infection during their stay (Appendix A). These patients, at emergency department arrival, exhibited altered mental status as determined by Glasgow Coma Score in high proportion, as well as hemodynamic disturbances illustrated by higher heart rate and systolic blood pressure. Routine laboratory markers such as C-reactive protein (CRP), procalcitonin (PCT) and ferritin showed higher levels in secondary-infected patients. Leucocyte cell counts in these patients were characterized by neutrophilia and higher neutrophil-to-lymphocyte ratio.

#### 3.2.2. Soluble Plasma Markers at Admission Associated with Secondary Infections

Inflammatory cytokines in plasma at hospital admission were higher in patients who developed secondary infection during their stay (Appendix A). Levels were similar to those observed in severe patients when they were classified according to O_2_ requirement but with additional cytokines such as IFNγ, IL-1β, IL-4 and IL-17A. Regarding soluble ICs, only sTim-3 showed increased levels in patients who subsequently became infected by another microorganism during the hospital stay. No differences in any of the studied thrombosis soluble markers were found.

#### 3.2.3. Immune Cell Subpopulations and Their Immune-Checkpoint Expression at Admission Associated with Secondary Infections

We compared the baseline immune subpopulations and the immune checkpoints expression between patients classified according to whether they experienced a secondary infection during they stay (Table 2). The monocytes from subsequently over-infected patients exhibited lower HLA-DR and higher PD-L1 expressions compared with no over-infected individuals. These patients also showed high expression of CD206 and CD226 in the neutrophils membrane (Table 2). As for DCs, it is worth pointing out that all the studied populations, myeloid, plasmacytoid, CD1c^+^ and CD141^+^, were decreased in patients who later underwent secondary infections. In these individuals, frequency of NK cells and both CD56^dim^ and CD56^bright^ subpopulations decreased and the expression of the activation receptor CD226 was diminished in these cells. Concerning the adaptive immune system, all analyzed T cells (CD3^+^, CD4^+^ and CD8^+^) revealed low expression of activation receptor CD226. Other alterations in T cells were the higher percentage of CD223^+^ and CD276^+^ cells. Both naïve and central memory γδ^+^ T cells had increased frequencies in secondary-infected patients. The B cells from these patients showed alterations with lower naïve and higher switched memory and plasma cells compared with the non-over-infected patients.

### 3.3. Immunological Profile Changes after Secondary Infections in COVID-19 Patients

Once it was established that COVID-19 patients who suffered an over-infection during their admission exhibited certain basal immunological alterations, we decided to study what immunological changes these patients suffered after they got a secondary infection. To do that, we compared all the previously studied immunological parameters and compared the previous levels (first and second samples) with the levels at the time of the positive culture in the thirty-three over-infected patients. We found that chemokine and cytokine levels decreased after the secondary infection (Table 3). In the same line, other soluble markers including the immune checkpoints, PD-1, Galectin-9 as well as P-selectin and its ligand (PSGL-1) were decreased. Total monocyte population (CD14^+^) increased after the secondary infection in accordance with an increase in intermediate (CD14^+^CD16^+^), and non-classical (CD14^dim^CD16^+^) subpopulations. At that moment, monocyte exhibited low expression of CD223 and GITR. Regarding membrane immune-checkpoints, NK cells exhibited general low expression of both activation (CD137, GITR) and inhibitory (V-domain Ig suppressor of T cell activation, VISTA) receptors. The CD4^+^ cells also showed a mixed pattern, with higher CD226 (activation) and Tim-3 (inhibitory) expression and lower GITR (activation), CD223 and B7H5 (inhibitory) expression after secondary infection (Table 3). It is worth noting that the expression of CD108, a receptor with Ig-like V-type domain with an unknown function in lymphocytes, is diminished in NK cells, CD4^+^ and CD8^+^ T cells. Other adaptive immune system alterations include an increase in central memory γδ^+^ T cells and change of naïve to transitional B cells.

### 3.4. Immunological Profile Changes after Corticosteroid Treatment in COVID-19 Patients

Finally, we proceeded to study the changes in immunological parameters due to COVID-19 treatments. The only medication that showed relative efficacy in alleviating COVID-19 symptoms during the first waves were the corticosteroids [2,32]. In our cohort, 18 patients were treated with corticosteroids (intravenous 8 mg or oral 6 mg of dexamethasone daily until exitus or discharge, mean treatment duration 4.7 days). In them, we analyzed the differences in the immune system variables in the samples immediately before and after treatment. We found a decrease in the chemokine CXCL10 after treatment (Table 4). Concerning immune cells, we found alterations on monocytes in several markers, increase in CD33, CD170 and CD223 and decrease in CD152, CD276 and HLA-DR. NK cells showed increase in B7H5, CD162 and CD270. T cells also showed B7H5 and CD270 increases after treatment accompanied by increases of CD108 in both CD4^+^ and CD8^+^ cells, CD226 in CD4^+^ and CD223 in CD8^+^. Other immune changes after corticosteroids included the increase in CD1c^+^ DCs, the decrease in total DCs and the unswitched subpopulation of B cells.

## 4. Discussion

Herein, we have described a high-dimensional comprehensive study of the baseline clinical and immunological differences in COVID-19 patients during the first wave of the pandemic in Madrid, Spain. We have defined potential early markers (upon arrival at ED) associated with both hospital requirement and one of the biggest complications in these patients, secondary infections [3,4,8]. In addition, thanks to the collection of longitudinal samples during the patients’ stays, we have sought to describe the immunological changes that occur due to secondary infection as well as after treatment with one of the most used drugs, corticosteroids.

We have found that at ED admission, COVID-19 patients showed differences in most of the immune populations studied when classified according to their outcome and O_2_ requirement during their hospital stay. For instance, monocyte population in severe patients (both exitus and OTI-required patients) were characterized by increment of myeloid-derived suppressor cells (m-MDSCs), higher expression of CD16, CD206 and PD-L1 and lower expression of HLA-DR. All these features are M2 hallmarks, suggesting an alternative polarization of monocytes in severe patients [23,33]. As for NK cells, in severe patients these cells showed increased expression of CD321, CD270 and GITR. The case of CD321, also known as junctional adhesion molecule 1 (JAM1) or F11R, is especially conspicuous. We included this receptor for having the Ig-like V-type domain of immune-checkpoints. It is known to play a major role in epithelial tight junctions as well as being a co-receptor for rotavirus entry [34,35]. However, its possible function in NK cells remains unknown.

Regarding the adaptive immune system, we found a high proportion of effector memory and terminally differentiated subpopulations, decreased CD226 expression and high CD162 expression in CD4^+^ T cells in severe patients. The CD162 (PSGL-1) has been described by having a role in inducing T cell exhaustion and driving effector and memory in T cells during acute infection [36,37,38]. The involvement of this receptor in the changes of T cell memory profile and the effect of its blockade in T cell function of COVID-19 patients should be further studied. In CD8^+^ cells, we found a similar profile to that observed in CD4^+^; however, it is striking that CD95 (Fas) is overexpressed in nearly all CD8^+^ subpopulations of severe COVID-19 patients. This molecule is a cell-surface receptor of the tumor necrosis factor superfamily, which has long been viewed as a death receptor that mediates apoptosis and could have a role in T cell apoptosis and lymphocytopenia observed in COVID-19 [39,40,41]. We must also remark that GITR expression showed increased levels, especially in CD8^+^ and NK cells from OTI required patients. This molecule could be modulating cytotoxic function in these cells [42,43].

Beyond the severe acute respiratory syndrome in COVID-19, a combination of virus- and drug-induced immunosuppression leads to sepsis and incidence of secondary infections, mainly in critically ill protracted patients [3,4,5,6,8,31]. Our study allows us to compare the clinical and immune alterations in these two phenomena at ED admission. Although we have observed that the basal profile of patients with O_2_ requirement shares features with the ones observed in patients with over-infection (higher inflammatory markers, cytokines and immune checkpoints), the profile is not completely the same. Clinically, we did not find differences in age; however, we found basal altered mental status, hemodynamic alterations and higher male frequency in secondary-infected patients. Moreover, we found the secondary-infected patients’ profiles were characterized by fewer alterations in immune checkpoint markers but exhibited greater increases in cytokines (e.g., IFNγ, IL-1β, IL-17A) as well as a higher numbers of alterations in DCs and B cells.

We have studied a large number of immune checkpoints, as well as candidate receptors for this role due to their having the Ig-like V-type domain [19,20]. We focused on these factors as they can not only be biomarkers but also therapeutic targets [25]. Some of the molecules analyzed here, such as CD321, GITR, CD226 or CD108, that have shown associations with different parameters of the patient’s evolution have been insufficiently studied in infectious diseases. For instance, this last receptor, CD108, also known as semaphorin 7A, has been described as having a role in modulating T cell response in a murine model of experimental autoimmune encephalomyelitis [44]; however, its role in human cells and its possible activity in infectious diseases is largely unknown. According to our data, its expression diminished after bacterial secondary infection and increased due to corticosteroid treatment in both CD4^+^ and CD8^+^ cells. The study of this receptor function in sepsis and septic shock would be an important field to deal with.

The CD226 (DNAX accessory molecule 1, DNAM-1) is expressed in both lymphoid (NK and T cells) and myeloid (neutrophil, monocytes and platelets) lineages [45,46]. This receptor is a costimulatory molecule responsible for activating the immune system. According to a recent work, a decrease in its expression may be caused by tumoral activity and CD226^+^ on CD8^+^ T cells correlated with improved progression-free survival following in melanoma [47]. The data we found for CD226 is curious, because critically ill patients showed low expression in both CD4^+^ and CD8^+^ T lymphocytes, while in the neutrophils of these same individuals its expression was increased. This fact leads us to think that, despite the ubiquitous expression of this receptor, different pathways depending on the cell lineage could mediate its transcriptional control. A possible mechanism involved could be by the transcription factor Eomes, as it is a key regulator of T cell maturation and it seems to be involved in the decrease in CD226 expression in CD8^+^ cells [48].

During the first wave of the pandemic, corticosteroids were one of the few treatments that showed efficacy in reducing 28-day mortality among those who were receiving either invasive mechanical ventilation or oxygen [1,2,32]. Regarding our data about immune changes to corticosteroids, it is noteworthy the expression of sialic acid-binding immunoglobulin-type lectins (Siglecs) on monocyte cells. Targeting Siglecs has recently emerged as a promising therapeutic strategy in cancer [49,50,51,52]. In our study, we found that expression of both Siglec-3 (CD33) and Siglec-5 (CD170) increased after corticosteroid treatment. These two receptors have recently been described as having immune checkpoint roles and usefulness as prognosis markers in different contexts [53,54,55,56]. Concerning lymphoid cells, we want to highlight the B7H5 (VISTA) increment in NK, CD8^+^ and CD4^+^ cells after corticosteroid treatment. Our data shows that the role of Siglecs and VISTA on the immunosuppressive effects of corticosteroid treatment should be explored in future studies. Finally, we must point out that we did not observe any expression alteration for any marker in neutrophils in any of the two longitudinal studies (neither before nor after secondary infection, nor after treatment with corticosteroids). This could be mainly due to the short half-life of these cells in circulation.

Despite the great advances in prophylaxis and treatment of SARS-CoV-2 infection, mortality and hospital requirements still represent a problem. Our study identified immune checkpoints and Ig-like V-type receptors in several cell populations by spectral cytometry associated with different features of COVID-19 physiopathology: O_2_ requirement, and secondary infections and corticosteroid treatment (Figure 1). Sometimes, most of these receptors follow the same pattern, similarly to what we found after the incidence of secondary infections, where both activators and inhibitors fell. Consequently, a multiparametric approach including a large number of markers in order to evaluate their balance is necessary [57]. The large number of markers herein provides clues as to which immune molecules/populations to include in subsequent studies, not only in COVID-19, but also in other pathologies.

## 5. Conclusions

Immune-checkpoints and Ig-like V-type receptors display expression disbalances according to severity and clinical course of COVID-19 patients.

## Figures and Tables

**Figure 1 jcm-11-03287-f001:**
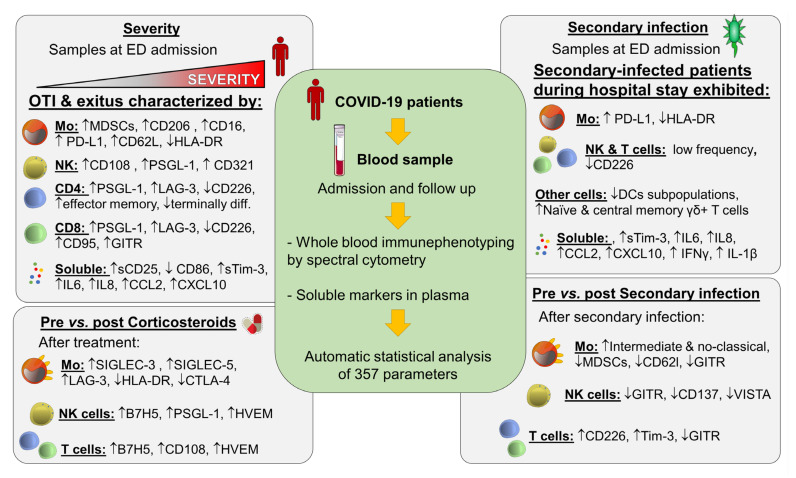
Graphical abstract: ED, Emergency Department; DCs, dendritic cells, MDSCs, myeloid-derived suppressor cells, Mo, monocyte; NK, natural killer; OTI, orotracheal-intubated patients; exitus, death patients.

**Table 1 jcm-11-03287-t001:** Immune subpopulations, immune checkpoints and activation markers’ expression at admission in COVID-19 patients classified according to their severity and final outcome.

Population		No Hospitalized (*n* = 22)	Hospitalized with No O_2_ (*n* = 10)	O_2_ Requirement (*n* = 23)	OrotrachealIntubation (*n* = 13)	*Exitus* (*n* = 19)	*p*-Value
Monocytes (MΦ)	m-MDSCs (% of total leukocytes)	0.95 (0.53–1.37)	0.68 (0.6–1.36)	0.74 (0.49–1.73)	2.61 (0.93–3.74)	1.55 (0.91–3.63)	0.04 *
	CD16	47,765 (42,068–51,544)	50,548 (47,680–51,714)	47,259 (41,927–56,258)	54,050 (48,199–72,597)	58,286 (50,114–68,235)	0.04 *
	CD137	12,375 (10,704–14,070)	15,033 (14,701–17,356)	14,905 (12,882–16,091)	15,963 (13,351–21,575)	14,529 (10,466–16,733)	0.03 *
	CD206	8009 (7246–8668)	8722 (7906–9937)	8492 (7200–9525)	9503 (8633–12,995)	9191 (7480–10,997)	0.03 *
	CD223	−2832 (−3599–(−2303))	−998 (−1743–2.1)	−1596 (−2967–(−446))	−1290 (−2342–1471)	−2497 (−3521–(−661))	0.02 *
	GITR	11,878 (8035–15,868)	17,396 (16,751–18,156)	16,795 (11,010–19,404)	19,646 (15,132–27,655)	16,033 (8000–19,872)	0.01 *
	HLA-DR	89,341 (67,377–119,767)	71,207 (58,657–111,503)	98,626 (53,345–154,812)	13,075 (116–34,984)	22,586 (3363–44,757)	<0.01 *
	HLA-DR (on classical MΦ)	41,211 (31,131–54,789)	32,889 (28,380–61,435)	44,596 (26,126–65,190)	17,781 (13,158–22,383)	25,505 (14,868–31,224)	<0.01 *
	PD-L1 (on classical MΦ)	19,373 (17,394–21,817)	19,117 (16,622–28,767)	22,188 (18,085–31,198)	27,525 (22,930–43,874)	26,049 (23,386–38,107)	0.01 *
	PD-L1 (on intermediate MΦ)	13,945 (9906–15,559)	12,795 (9162–16,247)	15,600 (13,303–19,005)	17,768 (14,871–20,041)	19,883 (15,799–24,578)	0.04 *
	CD62L (on classical MΦ)	3407 (2024–4631)	4498 (3424–5702)	4554 (2798–6036)	3830 (2057–4506)	5607 (4083–8203)	0.048 *
	CD62L (on intermediate MΦ)	2980 (1533–4121)	3931 (2519–5148)	3901 (3036–5636)	2563 (1444–3039)	4772 (3728–7378)	<0.01 *
	CD62L (on non-classic MΦ)	1402 (−383–3843)	6089 (4960–9107)	4895 (1090–6778)	486 (−118–3492)	5295 (2142–6421)	<0.01 *
Neutrophils	CD206	4754 (3979–4982)	6591 (5695–7236)	5698 (4311–7042)	5441 (4881–8097)	6266 (5119–8150)	0.02 *
	CD226	9314 (7735–10,897)	8814 (5757–10,431)	8893 (7126–10,495)	10,498 (8864–13,827)	11,449 (9576–11,957)	0.04 *
	Tim-3	−268 (−454–(−123))	96 (−248–765)	−56 (−160–280)	−157 (−297–72)	−76 (387–54)	0.03 *
Dendritic Cells (% of total leukocytes)	Myeloid (CD11c^+^CD123^+^)	0.23 (0.13–0.34)	0.18 (0.07–0.28)	0.15 (0.09–0.32)	0.047 (0.018–0.11)	0.09 (0.01–0.16)	0.01 *
	CD141^+^	0.074 (0.017–0.185)	0.031 (0.01–0.08)	0.05 (0.025–0.07)	0.013 (0.003–0.017)	0.02 (0.007–0.05)	0.02 *
NK cells	CD56^bright^ (% of total leukocytes)	0.15 (0.09–0.27)	0.11 (0.01–0.14)	0.09 (0.068–0.12)	0.04 (0.02–0.09)	0.05 (0.02–0.09)	0.01 *
	CD108	1001 (757–1122)	1409 (1211–1761)	1450 (993–1637)	1509 (1123–2554)	1343 (966–2045)	0.02 *
	CD137	5732 (4970–6693)	7283 (7102–8826)	7138 (6064–7936)	7487 (6121–10,070)	7043 (5112–7680)	0.03 *
	CD162 (PSGL-1)	5621 (4849–7189)	6191 (2458–10,102)	5915 (4413–8319)	13,321 (9722–16,325)	7689 (5892–11,417)	<0.01 *
	CD270 (HVEM)	37,940 (32,249–41,290)	41,220 (40,686–42,794)	41,617 (40,485–44,314)	40,851 (36,233–47,783)	41,951 (37,003–48,630)	0.03 *
	CD321	13,624 (12,363–14,122)	14,573 (13,506–19,158)	15,719 (14,455–18,211)	13,842 (12,854–22,062)	16,248 (14,482–21,413)	0.03 *
	GITR	6324 (3639–7357)	8786 (8697–9578)	8766 (4694–9623)	8949 (6872–13,358)	8600 (4009–11,205)	0.01 *
	GITR-L	7141 (5566–9358)	10,977 (10,013–11,386)	10,868 (8042–11,318)	9536 (7434–11,751)	8277 (6766–10,993)	0.02 *
	TIGIT	−1980 (−3328–(−466))	1233 (960–1897)	251 (−2306–1453)	−703 (−1798–718)	−38 (−3501–2315)	0.04 *
CD3^+^ lymphocytes	CD3^+^ (% on total leukocytes)	17.1 (14–21.8)	9.2 (3.1–21.5)	12.7 (5.2–17.2)	5.7 (2.3–8.9)	6.1 (2.7–13.1)	0.01 *
	CD108	1036 (791–1109)	1337 (1106–1523)	1335 (871–1514)	1439 (1015–2060)	1136 (925–1404)	0.03 *
	CD137	5380 (4631–5938)	6480 (6183–7581)	6397 (4622–6827)	6211 (5608–8997)	5970 (4605–6656)	0.048 *
	CD152 (%)	1.35 (0.95–3.3)	6.00 (2.9–6.6)	3.44 (1.46–4.84)	2.43 (0.75–3.95)	4.44 (1.77–7.19)	0.04 *
	CD162 (PSGL-1)	5404 (4793–6388)	6445 (6022–8215)	6027 (4362–7906)	10,556 (8427–12,860)	6719 (5076–11,404)	<0.01 *
	CD223	−1041 (−1319–(−835))	−210 (−427–259)	−262 (−819–96)	−202 (−855–638)	−581 (−1108–218)	0.01 *
	CD223 (%)	1.23 (0.48–2.37)	5.40 (1.36–8.8)	4.19 (2.3–7.82)	14.44 (6.77–37.4)	6.41 (1.64–20.8)	<0.01 *
	CD226 (%)	62.20 (48–78)	41.20 (38.5–68.6)	49.75 (25.1–67.6)	54.7 (47.6–66.7)	12.0 (6–64)	0.02 *
	GITR	5756 (3682–6965)	7899 (7509–8308)	7752 (4192–8869)	7833 (6700–12,074)	7687 (3707–9623)	0.04 *
	TIGIT	−1805 (−3119–(−688))	340 (69–927)	24 (−2422–876)	−585 (−1538–51)	−1307 (−3652–952)	0.04 *
	Tim-3	−36 (−157–52)	93 (59–154)	77 (−11–155)	58 (−96–94)	53 (−32–133)	0.03 *
CD4^+^ T cells	CD4^+^ (% on CD3^+^)	41 (31–52)	40 (32–46)	40 (27–50)	27 (24–2−34)	29 (18–38)	0.03 *
	CD4em	12.5 (6.8–52.2)	11.7 (6.9–26.9)	26 (13–34.5)	8.8 (6.5–13.7)	29.3 (20–53)	0.04 *
	CD4td	3.3 (1–14.8)	3.6 (1.7–9.2)	4.6 (1.5–17.8)	7.5 (5.2–14.9)	18.3 (9.4–22.4)	0.01 *
	CD95	28,072 (25,207–32,591)	38,858 (25,782–52,147)	39,334 (26,476–48,056)	25,126 (23,151–30,421)	34,386 (32,863–43,106)	0.04 *
	CD95 on CD4n	10,044 (7857–12,140)	14,434 (11,777–19,201)	13,791 (8851–19,808)	8366 (7302–10,149)	13,444 (11,585–19,296)	0.02 *
	CD108	1044 (777–1091)	1341 (1138–1554)	1359 (882–1539)	1343 (1028–2195)	1171 (915–1530)	0.03 *
	CD108 (%)	64.70 (56–69)	53.80 (47.4–62.7)	56.5 (51–64.2)	65.3 (57.7–72.2)	68.85 (53.7–76)	0.04 *
	CD162 (PSGL-1)	5444 (4809–6069)	6299 (5553–7911)	6371 (4414–8705)	9790 (8489–13,163)	6975 (5501–9656)	<0.01 *
	CD223	−1026 (−1299–(−814))	−122 (−330–391)	−208 (−853–131)	−233 (−813–828)	−563 (−1019–267)	0.01 *
	CD223 (%)	1.94 (1.32–5.1)	10.50 (4.37–17.9)	7.45 (2.67–16.5)	16.4 (9.14–47.1)	14.8 (4.07–34.6)	<0.01 *
	CD226 (%)	66.20 (44.5–78.7)	43.30 (38.9–69)	49.5 (28.47–67.5)	56.6 (44.17–63)	26.3 (19.32–54.3)	0.048 *
	GITR-L	6152 (4177–7700)	8802 (8358–9421)	8696 (6227–9672)	7276 (5723–9199)	7040 (5452–8249)	0.047 *
	TIGIT	−1869 (−3057–(−781))	432 (216–1224)	176 (−2448–992)	−547 (−1451–312)	−1335 (−3623–975)	0.02 *
	Tim-3	−33 (−150–63)	133 (92–181)	105 (−2–170)	62 (−90–101)	51 (−33–156)	0.01 *
CD8^+^ T cells	CD95	28,294 (24,290–32,358)	34,163 (30,558–42,046)	35,421 (28,778–44,111)	28,917 (25,500–31,972)	38,506 (34,680–47,282)	0.01 *
	CD95 on CD8em	62,501 (59,155–65,520)	64,776 (64,905–91,663)	67,625 (61,148–78,483)	66,515 (61,442–72,600)	73,030 (67,485–76,648)	0.03 *
	CD95 on CD8n	14,040 (11,774–17,343)	15,616 (13,952–16,373)	19,502 (15,790–27,431)	14,319 (10,584–16,957)	22,946 (17,821–30,006)	0.01 *
	CD95 on CD8td	43,095 (40,524–45,710)	44,195 (40,622–46,163)	43,763 (42,306–51,144)	44,161 (41,126–49,327)	48,633 (46,499–51,618)	0.04 *
	CD137	5572 (4914–5907)	6179 (5981–7071)	6103 (4956–6422)	6653 (5460–9098)	6077 (5001–6970)	0.04 *
	CD162 (PSGL-1)	6126 (4876–7704)	8503 (6556–9800)	6690 (4427–8301)	11,804 (9267–14,075)	7002 (4975–13,726)	<0.01 *
CD8^+^ T cells	CD223	−1201 (−1316–(−942))	−373 (−643–83)	−310 (−988–(−47))	−304 (−954–424)	−711 (−1226–107)	0.01 *
	CD223 (%)	5.63 (2.88–7.8)	10.8 (3.8–24.8)	8.16 (4.41–22.3)	22.35 (13.2–41.4)	14.5 (6.6–34.2)	0.01 *
	CD226 (%)	65.1 (48.75–69.1)	55.40 (43.6–77.4)	68.35 (52.45–77.3)	57.8 (44.3–68.6)	33.65 (10–58.1)	0.03 *
	GITR	5831 (3668–6673)	7434 (6981–7621)	7200 (4322–8030)	8038 (6458–12,341)	7256 (3877–9892)	0.01 *
	Tim-3	−43 (−174–33)	89 (77–160)	99 (20–150)	40 (−112–70)	49 (−44–135)	0.01 *
γδ^+^ T cells	γδ^+^ (on total leukocytes)	0.35 (0.18–0.75)	0.08 (0.06–0.12)	0.22 (0.06–0.51)	0.14 (0.05–0.32)	0.09 (0.06–0.21)	0.04 *
B cells	CD19^+^ (% on total leukocytes)	2.61 (1.37–4.38)	1.52 (0.92–3.21)	1.90 (0.88–2.5)	0.72 (0.48–1.52)	0.69 (0.15–1.71)	<0.01 *
	trB cell (% on total leukocytes)	1.04 (0.27–1.39)	0.28 (0.18–0.38)	0.40 (0.16–0.68)	0.34 (0.15–0.69)	0.20 (0.05–0.72)	0.02 *
	Plasma cell (% on CD19^+^)	9.15 (2.62–14.4)	2.31 (1.9–4.16)	4.02 (1.22–10.5)	21.5 (7–24.85)	10.1 (3.12–16.5)	0.04 *

Data are percentages or mean intensity of fluorescence in biexponential scale expressed as Median (IQR). * Kruskal–Wallis test. m-MDSCs, monocyte myeloid-derived suppressor cells; CD4em, CD4^+^ effector memory T cells defined as CD45RA-CD27^−^; CD4td, CD4^+^ terminally differentiated T cells defined as CD45RA^+^CD27^−^; CD4n, CD4^+^ naïve T cells defined as CD45RA^+^CD27^+^; CD8em, CD8^+^ effector memory T cells defined as CD45RA-CD27^−^; CD8td, CD8^+^ terminally differentiated T cells defined as CD45RA^+^CD27^−^; CD8n, CD8^+^ naïve T cells defined as CD45RA^+^CD27^+^; trB cell, transitional B cell defined as CD19^+^IgD^dim^CD38^+^.

**Table 2 jcm-11-03287-t002:** Immune subpopulations, immune checkpoints and activation markers expression at admission in COVID-19 patients classified according to whether they developed a secondary infection during their hospital stay.

Type		No Secondary Infection (*n* = 54)	Secondary Infection (*n* = 33)	*p*-Value
Monocytes	HLA-DR	74,440 (45,851–111,503)	38,090 (13,075–89,843)	0.048 *
	PD-L1	14,218 (11,665–18,379)	17,496 (15,090–21,905)	0.03 *
Neutrophils	CD206	4875 (4104–6600)	6244 (5094–8127)	0.03 *
	CD226	8805 (7398–10,773)	10,708 (9325–11,952)	0.01 *
Dendritic cells (% of total leukocytes)	Myeloid (CD11c^+^CD123^+^)	0.175 (0.092–0.315)	0.073 (0.018–0.165)	<0.01 *
	Plasmacytoid (CD11c-CD123^+^)	0.044 (0.02–0.114)	0.011 (0.005–0.064)	0.01 *
	CD1c^+^	0.180 (0.076–0.418)	0.077 (0.024–0.18)	<0.01 *
	CD141^+^	0.048 (0.017–0.084)	0.010 (0.003–0.031)	<0.01 *
NK cells	CD3-CD56^+^ (% on total leukocytes)	2.905 (1.39–4.67)	0.770 (0.475–1.895)	<0.01 *
	CD56^dim^ (% of total NKs)	2.670 (1.29–4.51)	0.720 (0.460–1.82)	<0.01 *
	CD56^brigh^t (% of total NKs)	0.105 (0.07–0.178)	0.045 (0.014–0.098)	<0.01 *
	CD226	10,064 (8033–12,322)	8733 (6803–10,029)	0.01 *
NKT cells	CD56^+^CD3^+^ (% on total leukocytes)	1.0 (0.5–2.3)	0.5 (0.2–1.0)	0.03 *
CD3^+^ lymphocytes	CD3^+^ (% on total leukocytes)	14.6 (6.4–19.8)	6.1 (2.8–10.3)	<0.01 *
	CD223 (%)	2.65 (0.84–8.01)	7.27 (2.38–20.2)	0.01 *
	CD226	9491 (6584–12,228)	7216 (5546–8445)	<0.01 *
	CD226 (%)	57.1 (41.4–72.7)	35.6 (11.05–67.1)	0.02 *
CD4^+^ T cells	CD4^+^ (% on total leukocytes)	38.3 (30.6–50.8)	27.4 (20.8–39.5)	<0.01 *
	CD4^+^ (% on CD3^+^)	63.3 (55.9–69.3)	49.2 (25.5–66.3)	0.01 *
	CD223 (%)	5.67 (1.94–18.1)	12.6 (5.75–20.65)	0.03 *
	CD226	9347 (6878–12,193)	7306 (5933–8640)	<0.01 *
	CD226 (%)	58.3 (38.9–77.2)	38.7 (18.05–60.55)	<0.01 *
	CD276 (%)	14.7 (6.2–26.3)	5.08 (1.43–22.8)	0.02 *
CD8^+^ T cells	CD223 (%)	6.84 (3.8–17.9)	15.40 (7.18–27.7)	0.03 *
	CD226	10,016 (7291–12,953)	7824 (6081–9337)	0.03 *
	CD226 (%)	62.9 (46.3–73.7)	43.6 (18.65–64.05)	0.01 *
	CD276 (%)	12.6 (7.08–22.6)	5.0 (2.57–17.65)	0.02 *
	CD95 on CD8em	64,506 (59,473–73,422)	72,949 (66,760–76,753)	0.01 *
γδ^+^ T cells	Naïve (%)	8.2 (3.1–28.2)	30.5 (9.9–53.6)	0.01 *
	Central memory (%)	16.6 (10.1–38.9)	33.1 (21.4–44.8)	0.01 *
B cells	CD19^+^ (% on total leukocytes)	1.9 (1.1–3.7)	0.8 (0.3–1.6)	<0.01 *
	nB cell (% on CD19^+^)	38.6 (22.2–51.2)	20.5 (3.3–38.8)	0.02 *
	smB cell (% on CD19^+^)	13.7 (6.6–32)	26.0 (14–48.8)	0.03 *
	Plasma cell (% on CD19^+^)	3.2 (0.7–6.8)	6.6 (1.1–12.6)	0.04 *

Data are percentages or mean intensity of fluorescence in biexponential scale expressed as median (IQR); * Mann–Whitney test. CD4em, CD4^+^ effector memory T cells defined as CD45RA^−^CD27^−^; CD4td, CD4^+^ terminally differentiated T cells defined as CD45RA^+^CD27^−^; CD4n, CD4^+^ naïve T cells defined as CD45RA^+^CD27^+^; CD8em, CD8^+^ effector memory T cells defined as CD45RA^−^CD27^−^; CD8td, CD8^+^ terminally differentiated T cells defined as CD45RA^+^CD27^−^; CD8n, CD8^+^ naïve T cells defined as CD45RA^+^CD27^+^; nB cell, naïve B cell defined as CD19^+^IgD^+^CD27^−^; smB cell, switched memory B cell defined as CD19^+^IgD^−^CD27^+^.

**Table 3 jcm-11-03287-t003:** Immune parameters and markers previous and after secondary infection in COVID-19 patients.

Type		Pre-Secondary Infection	Post-Secondary Infection	*p*-Value
Chemokines and Cytokines	CXCL10	154.9 (98.5–409)	124.2 (66–182)	0.01 *
	IFNγ	30.4 (5.9–44.4)	1.4 (0–38.8)	0.01 *
	Il-2	3.7 (0–8)	0.7 (0–5)	0.046 *
	IL-6	78.1 (26.4–230)	44 (5.2–91)	0.01 *
	IL-10	8 (0–14.5)	1.8 (0–10.2)	0.03 *
	TNFα	0 (0–5.3)	0 (0–0)	<0.01 *
Immune checkpoint	sPD-1	4.6 (3.4–14.1)	3.6 (2–7.2)	<0.01 *
	Galectin-9	110,810 (39,726–162,758)	44,249 (33,744–68,212)	<0.01 *
Thrombosis	sP-Selectin	14,173 (6786–39,637)	8819 (1326–24,858)	0.03 *
	sPSGL-1	206.4 (0–11,689)	0 (0–266)	0.01 *
Monocytes	CD14^+^ (% of total leukocytes)	5.0 (1.8–8.6)	6.8 (4.4–9.1)	0.04 *
	m-MDSCs (% of total leukocytes)	1.18 (0.71–2.3)	0.87 (0.52–1.6)	0.02 *
	CD223	−575.9 (−1338–1487)	−1568 (−2472–1186)	0.04 *
	GITR	18,059 (9591–23,306)	10,086 (6968–21,470)	0.03 *
	Intermediate MΦ (% of total leukocytes)	0.2 (0.1–0.3)	0.3 (0.2–0.5)	0.01 *
	Non-classic MΦ (% of total leukocytes)	0.2 (0.1–0.5)	0.7 (0.4–1)	**<0.01 ***
	CD62L (on non-classic MΦ)	3759 (270–7275)	912 (−6.6–2196)	0.01 *
NK cells	B7H5	2715 (2120–3783)	2020 (1620–3566)	0.04 *
	CD108	1661 (1179–2107)	863 (494–1995)	<0.01 *
	CD137	7491 (4999–9334)	5274 (4089–8188)	0.02 *
	GITR	8974 (4634–13,211)	4495 (3643–10,609)	<0.01 *
	GITR-L	10,434 (8021–11,657)	7648 (6541–11,361)	0.01 *
CD4^+^ T cells	CD4cm	34 (13.2–43.3)	45 (30–57.7)	**<0.01 ***
	CD4td	15.8 (6.1–30.2)	3.2 (1.7–6.8)	**<0.01 ***
	B7H5	2284 (1941–3077)	1721 (1396–2693)	0.047 *
	CD108	1459 (1030–1659)	896 (653–1857)	0.01 *
	CD223	99.8 (−565–1429)	−342 (−895–1128)	0.03 *
	CD226 (%)	46.7 (31–61.6)	63.2 (48–78)	<0.01 *
	CD226	7943 (6665–10,849)	10,158 (7808–13,161)	0.02 *
	GITR	8249 (4171–13,928)	4357 (3587–9626)	0.02 *
	Tim-3 (%)	43.2 (37.4–50.6)	50.85 (39–61.3)	0.02 *
CD8^+^ T cells	CD108	1252 (924–1776)	956 (711–1604)	0.02 *
	CD137	6311 (4754–8189)	4843 (3938–7123)	0.03 *
	CD152 (%)	21.2 (9.2–33.4)	36.2 (16.6–48.8)	0.02 *
	GITR	7572 (4274–11,453)	4130 (3229.8–8635)	<0.01 *
γδ^+^ T cells	γδ^+^ (% on total leukocytes)	0.135 (0.04–0.365)	0.22 (0.11–0.49)	0.04 *
	Central memory (%)	28.55 (14.3–46.1)	42 (29–51.6)	0.01 *
B cells	nB cell (% on CD19^+^)	28.15 (17.6–51)	18.8 (6.5–33)	0.02 *
	trB cell (% on CD19^+^)	7.78 (0.86–12.7)	12.8 (4.2–20.5)	0.02 *
	CD27^−^IgD^−^ (% on CD19^+^)	24.25 (15–39.8)	35.8 (18.6–53)	0.04 *

Data are percentages or mean intensity of fluorescence in biexponential scale expressed as median (IQR); * Mann–Whitney test. In bold, those *p*-values with high significance (defined by *p*-value < FDR by Benjamini–Hochberg method). m-MDSCs, monocyte myeloid-derived suppressor cells; CD4cm, CD4^+^ central memory T cells defined as CD45RA-CD27-; CD4td, CD4^+^ terminally differentiated T cells defined as CD45RA^+^CD27-; nB cell, naïve B cell defined as CD19^+^IgD^+^CD27-; trB cell, transitional B cell defined as CD19^+^IgD^dim^CD38^+^.

**Table 4 jcm-11-03287-t004:** Immune parameters and markers before and after treatment with corticosteroids in COVID-19 patients.

**Type**		**Pre-Treatment**	**Post-Treatment**	** *p* ** **-Value**
Chemokines	CXCL10	232 (106–693)	138 (73–234)	0.03
Monocytes	CD33 (Siglec-3)	15,857 (7479–18,207)	19,236 (14,302–24,540)	0.02
	CD152 (CTLA-4)	639 (314–1259)	−452 (−2574–712)	<0.01
	CD170 (Siglec-5)	−2215 (−3086–(−714))	446 (−1662–7808)	0.01
	CD223	−1495 (−3479–(−746))	−576 (−2338–2259)	0.01
	CD276 (B7H3)	3225 (2268–4559)	625 (−2643–4110)	0.02
	HLA-DR	103,485 (64,040–111,089)	55,363 (24,152–93,803)	0.047
	HLA-DR (on classical MΦ)	48,135 (33,733–68,059)	31,843 (17,650–47,653)	0.02
	PD-L1 (on non-classic MΦ)	16,881 (4664–27,670)	33,213 (6880–40,379)	0.04
Dendritic Cells	CD1c^+^ (% of total DC)	17.5 (11.4–23.1)	6.8 (2.53–14.53)	0.02
NK cells	B7H5	2223 (1780–2530)	2769 (1983–4710)	0.01
	CD162 (PSGL-1)	6178 (4079–9165)	8716 (5885–11,483)	0.04
	CD270 (HVEM)	40,659 (33,705–42,478)	42,504 (39,611–56,296)	0.02
CD4^+^ T cells	B7H5	1857 (1629–2030)	2273 (1746–4452)	0.01
	CD108	1085 (789–1289)	1431 (866–2706)	0.03
	CD226	7179 (6213–9084)	9551 (7460–12,402)	0.03
	CD270 (HVEM)	33,929 (29,624–34,958)	37,145 (34,469–49,570)	<0.01
CD8^+^ T cells	B7H5	1932 (1746–2181)	2397 (1924–3829)	0.01
	CD108	970 (848–1109)	1250 (915–2286)	0.02
	CD223 (%)	5.11 (3–18.1)	17 (5.06–69.3)	0.04
	CD270 (HVEM)	35,512 (33,183–36,840)	38,963 (35,854–47,186)	<0.01
B cells	usB cell (% on CD19^+^)	4.16 (2.35–6.35)	7.7 (3.76–13.7)	0.047

Data are mean intensity of fluorescence in biexponential scale expressed as median (IQR); * Mann–Whitney test. DCs, dendritic cells; usB cell, unswitched B cell defined as CD19^+^IgD^+^CD27^+^.

## Data Availability

The data that support the findings of this study are available from the corresponding authors upon reasonable request.

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
