# Peer review of "Differential Immune Checkpoint and Ig-like V-Type Receptor Profiles in COVID-19: Associations with Severity and Treatment"

_jcm, 2022, doi:10.3390/jcm11123287_

Round 1

Reviewer 1 Report

The title of the article is too long and not attractive enough

Be more careful in choosing keywords and use related words.

There is no mention of the consent form for obtaining samples from patients.

attention to using acronyms, for example, explain acronyms for the first time and subsequently use the acronym in the context.

In order to prevent the repetition of words "at admission associated to COVID-19 severity" was rewrite in line 174 and deleted in all subdivided titles

what means of qSOFA in line 179

LINE: 181, C-reactive protein=CRP 

LINE 183: Which of the 5 groups of Covid-19 patients is considered as severe patients?

LINE 193: virgule is need after ICs

LINEuse -....- instead of virgule

TABLE 1: The difference between the five groups was compared to which group?

LINE 229: Correct the spelling of infection 

line 230: hospitalization instead of stay

line 234: culture of samples such as blood, urine, BALF and STOOL

LINE 235: What is the full name of BALF?

line 261: what is the full name of tim-3?

totally, The results obtained are not clear among the 5 defined groups

line 311: The duration of the drug is not mentioned

Author Response

The title of the article is too long and not attractive enough

We thank the reviewer for his/her comment. Following the reviewer suggestion, we have changed the title.

Be more careful in choosing keywords and use related words

Following this recommendation, we have changed some of the keywords in our manuscript.

There is no mention of the consent form for obtaining samples from patients. (line 117)

In the “2.1. Patient recruitment and study design” section we stated all patient signed an informed consent:

Line 131-132 “Participants signed an informed consent and data were anonymized before study inclusion.”

 attention to using acronyms, for example, explain acronyms for the first time and subsequently use the acronym in the context.

We see eye to eye with this commentary. In the current version we have revised all the acronyms and corrected some vagueness.

In order to prevent the repetition of words "at admission associated to COVID-19 severity" was rewrite in line 174 and deleted in all subdivided titles

We have deleted “at admission…” in subdivided titles in order to avoid repetitions.

what means of qSOFA in line 179

qSOFA means quick-Sequential Organ Failure Assessment. In the current version it has been included.

LINE: 181, C-reactive protein=CRP 

In the current version it has been included.

LINE 183: Which of the 5 groups of Covid-19 patients is considered as severe patients?

In the current version we have corrected this vagueness. Note that we have described severe patients as exitus and orotracheal intubated patients.

LINE 193: virgule is need after ICs

In the current version we have corrected this vagueness.

LINEuse -....- instead of virgule

In the current version we have corrected this vagueness.

TABLE 1: The difference between the five groups was compared to which group?

The statistical analysis in the five groups comparison consisted in ANOVA (Kruskal-Wallis) with no reference group.

LINE 229: Correct the spelling of infection 

The word infection has been corrected.

line 230: hospitalization instead of stay

Following the recommendation, we have changed the word.

line 234: culture of samples such as blood, urine, BALF and STOOL

We have rephrased it according to the reviewer’s suggestion.

LINE 235: What is the full name of BALF?

            BALF stand for bronchoalveolar lavage fluid. This vagueness has been corrected.

line 261: what is the full name of tim-3?

Tim-3 stand for T cell immunoglobulin and mucin-domain containing-3. This vagueness has been corrected.

totally, The results obtained are not clear among the 5 defined groups

We have included the numeric data as Median and IQR. We consider these is the proper manner to describe the data distribution and compare them among the groups.

line 311: The duration of the drug is not mentioned

In the current version, we have included the mean treatment duration (4.7 days).

Reviewer 2 Report

This is an important contribution the scientific community.

I can follow the article in the main text. Unfortunately, I could not access the supplementary material even though I asked the publisher for. So, I have little information on the age distribution of the individuals to give an example.

What I miss is an extended analysis of B cells including the age - associated B cells referenced below. This may be very important for elderlies more susceptible to COVID-19 infection.

As this is a very unique manuscript in its richness of information, the antigen - specific antibody response differentiating the four IgG isotypes and IgA will be important as B cells and antibodies are critical (or negatively supportive) for the defense against the virus.

1.         Nipper AJ, Smithey MJ, Shah RC, Canaday DH, Landay AL. 2018. Diminished antibody response to influenza vaccination is characterized by expansion of an age-associated B-cell population with low PAX5. Clin Immunol 193:80-87.

2.         Cancro MP. 2020. Age-Associated B Cells. Annu Rev Immunol 38:315-340.

3.         Song W, Antao OQ, Condiff E, Sanchez GM, Chernova I, Zembrzuski K, Steach H, Rubtsova K, Angeletti D, Lemenze A, Laidlaw BJ, Craft J, Weinstein JS. 2022. Development of Tbet- and CD11c-expressing B cells in a viral infection requires T follicular helper cells outside of germinal centers. Immunity 55:290-307 e5.

4.         Mouat IC, Horwitz MS. 2022. Age-associated B cells in viral infection. PLoS Pathog 18:e1010297.

Author Response

This is an important contribution the scientific community.

            We thank the reviewer for his/her kind words.

I can follow the article in the main text. Unfortunately, I could not access the supplementary material even though I asked the publisher for. So, I have little information on the age distribution of the individuals to give an example.

We greatly regret that you were unable to access the supplementary information. The information contained therein helps to better understand the manuscript (e.g. Supplementary Table 1 described all the variables included in our study such as the B cells subpopulations defined in one of the comments below).

We have re-attached the supplementary material and we hope that this time you will be able to access it.

What I miss is an extended analysis of B cells including the age - associated B cells referenced below. This may be very important for elderlies more susceptible to COVID-19 infection.

As described in Supplementary Table 1, beyond B cell main population (CD19+CD24+CD56-HLA-DR+) we have analyzed 8 different B cells subpopulations included:

  • Total memory B cells (HLA-DR+CD27+)
  • Plasma B Cells (IgD-CD38highCD27+)
  • Transitional B cells (IgDdimCD38+)
  • Naïve (IgD+CD27-) B cells
  • Switch (IgD-CD27+) B cells
  • Unswitch (IgD+CD27+) B cells
  • CD27-IgD- cells

Regretfully we did not include the Age-associated B cell population as the T-bet marker. However, we consider that our 8 analyzed subpopulations are informative enough for an appropriate description of B cell phenotype.

As this is a very unique manuscript in its richness of information, the antigen - specific antibody response differentiating the four IgG isotypes and IgA will be important as B cells and antibodies are critical (or negatively supportive) for the defense against the virus.

Following the reviewer’s recommendation, we have analyzed IgA and IgG specific antibodies against SARS-CoV-2 Spike RBD domain and Spike S1 domain in the patients’ plasma (please see figures R1 and R2 below). Note that, we have not found differences in levels between disease severity groups as some of the patients has low or undetectable amounts. We would like to remark we recruited the patients during the first two waves of COVID-19 in Spain (April 26, 2020 to November 20, 2020) before massive vaccination campaign. In addition, patients arrived to emergency department at early stages of infection course probably before they developed humoral response.

Figure R1. Antibody IgA levels in patients’ plasma at emergency department admission. Levels (ng/mL) of specific anti-SARS-CoV-2 RBD domain IgA (A) and anti-SARS-CoV-2 S1 domain IgA (B) in patients’ plasma are shown. Levels were determined by using LEGENDplex™ SARS-CoV-2 Serological IgA Panel following manufacturers’ protocol. ANOVA Kruskal-Wallis test was used for statistical analysis. K.W., Kruskal Wallis statistic; n.s., non significant.

Figure R2. Antibody IgG levels in patients’ plasma at emergency department admission. Levels (ng/mL) of specific anti-SARS-CoV-2 RBD domain IgG (A) and anti-SARS-CoV-2 S1 domain IgG (B) in patients’ plasma are shown. Levels were determined by using LEGENDplex™ SARS-CoV-2 Serological IgG Panel following manufacturers’ protocol. ANOVA Kruskal-Wallis test was used for statistical analysis. K.W., Kruskal Wallis statistic; n.s., non significant.

Round 2

Reviewer 2 Report

This is a careful analysis.

The answers to my questions were adequate. 

This manuscript is a resubmission of an earlier submission. The following is a list of the peer review reports and author responses from that submission.